# Time Trends of Crohn’s Disease in Catalonia from 2011 to 2017. Increasing Use of Biologics Correlates with a Reduced Need for Surgery

**DOI:** 10.3390/jcm9092896

**Published:** 2020-09-08

**Authors:** Eduard Brunet, Emili Vela, Luigi Melcarne, Montserrat Clèries, Caridad Pontes, Laura Patricia Llovet, Pilar García-Iglesias, Marta Gallach, Albert Villòria, Mercedes Vergara, Xavier Calvet

**Affiliations:** 1Servei Aparell Digestiu, Hospital Universitari Parc Taulí, 08208 Sabadell, Spain; eduardbrunet91@gmail.com (E.B.); lmelcarne@tauli.cat (L.M.); lpllovet@tauli.cat (L.P.L.); Pgarciai@tauli.cat (P.G.-I.); martagmgm@gmail.com (M.G.); avilloria@tauli.cat (A.V.); mvergara@tauli.cat (M.V.); 2Departament de Medicina, Universitat Autònoma de Barcelona, 08193 Bellaterra, Spain; 3Unitat d’Informació i Coneixement, Servei Català de la Salut, Generalitat de Catalunya, 08002 Barcelona, Spain; evela@catsalut.cat (E.V.); mcleries@catsalut.cat (M.C.); 4Àrea del Medicament, Servei Català de la Salut, 08002 Barcelona, Spain; caridad.pontes@uab.cat; 5Departament de Farmacologia, de Terapèutica i de Toxicologia, Universitat Autònoma de Barcelona, 08193 Bellaterra, Spain; 6CIBERehd Instituto de Salud Carlos III, 28029 Madrid, Spain

**Keywords:** epidemiology, inflammatory bowel disease, Crohn’s disease

## Abstract

Background and Aims: Data from clinical trials suggest that biological drugs may improve the outcomes in Crohn’s disease (CD) by reducing the need for surgery or hospitalization. The aim of this study is to evaluate the time-trends of the use of biological drugs and other treatments for CD, and its relationship with outcomes in Catalonia. Materials and Methods: All patients with CD included in the Catalan Health Surveillance System (containing data on a population of more than 7.5 million) from 2011 to 2017 were identified. The exposures to different treatments for inflammatory bowel disease were retrieved from electronic invoicing records. Results: Between 2011 and 2017, the use of salicylates, corticosteroids and immunosuppressive treatment fell from 28.8% to 17.1%, 15.8% to 13.7%, and 32.9% to 29.6%, respectively (*p* < 0.001). Biological treatment use rose from 15.0% to 18.7% (*p* < 0.001). Ostomy rates per 1000 patients/year fell from 13.2 in 2011 to 9.8 in 2017 (*p* = 0.003), and surgical resection rates from 24.1 to 18.0 (*p* < 0.001). The rate of CD-related hospitalizations per 1000 patients/year also fell, from 92.7 to 72.2 (*p* < 0.001). Conclusions: Biological drug use rose from 15.0% to 18.7% between 2011 and 2017. During this period, we observed an improvement in the outcomes of CD patients.

## 1. Introduction

Crohn’s disease (CD) is an inflammatory bowel disease (IBD) that causes gastrointestinal tract inflammation [1]. Its prevalence and incidence are increasing: currently, its prevalence ranges between 28 and 322 per 100,000 inhabitants in Western Europe. The incidence and prevalence are especially high in Western Europe and North America [2,3,4].

CD has a variable clinical course, with the alternation of periods of remission and flares of active disease. The annual hospitalization rate approaches 20%; furthermore, about 50% of patients require surgery within 10 years of diagnosis [5]. Biological treatment (anti-tumor necrosis factor, anti-integrin and anti-interleukin 12/23) has changed the management of IBD, improving clinical and endoscopic remission rates [6] and reducing the need for surgery in clinical trials. The subanalysis of the SONIC trial suggested that the use of infliximab plus azathioprine correlated with fewer disease complications and fewer surgical procedures [7]. The ACCENT I trial and the CHARM trial showed a reduction in surgical procedures in the groups treated with infliximab and with adalimumab when compared to the groups receiving placebo [8,9]. Finally, Sandborn et al. recently showed that treatment with ustekinumab reduced surgery rates at two years by 30%–50% compared to placebo [10]. However, the data on the effects of biological drugs on outcomes in clinical practice are limited and controversial [11,12,13,14,15].

The primary aim of the present study was to trace the time-trends of the use of different CD treatments (and especially biologics) in Catalonia from 2011 to 2017 and to evaluate its correlation with disease outcomes such as surgery and hospitalization.

## 2. Material and Methods

All patients with CD included in the Catalan Health Surveillance System (CHSS) [16] between 2011 and 2017 were identified, according to their ICD-9-CM codes (the list of ICD-9 codes used is appended as Appendix A). The CHSS is a population-based healthcare register including data on more than 7.5 million individuals. The CHSS database has a unique personal identification number. The database and the baseline patients’ data have been described in detail in previous articles [17].

The exposures to different IBD treatments were retrieved from the electronic invoicing records for the same period. IBD treatments included biological drugs (infliximab, adalimumab, golimumab, vedolizumab and ustekinumab), immunosuppressive agents (azathioprine, 6-mercaptopurine and methotrexate), corticosteroids and salicylates. A patient was considered to have received a particular biological drug during a specific year if at least one prescription of the biologic was dispensed during this period. The list of active principles of the treatments studied is appended as Appendix A.

Data on the number of surgical procedures and the hospitalization rate of CD patients (scheduled and unscheduled) were also obtained from the CHSS and expressed both per 1000 patients/year and per 100,000 inhabitants. All-cause hospitalization was analyzed. Hospitalization related to CD, infections or neoplasms was also evaluated. ICD-9-CM codes for the different surgical procedures are detailed in Appendix A.

### 2.1. Statistical Methods

To calculate the annual rate per patient/year for surgical procedures and for the drugs used, we estimated the time at risk for each patient in each of the periods. Exposure periods were initiated on January 1st (or at the date of diagnosis for incident patients) and ended on December 31st of each year (or with the death of the patient). The sum of the drugs invoiced in the period was used as a numerator. The denominator was the number of patients/year. The statistical significance of the global variation of rates during this time period was calculated using generalized linear models. Pearson correlation coefficients were calculated between the rates of performing surgical procedures and the percentages of patients treated with each group of drugs. A *p* value of 0.05 or lower was considered as significant. The statistical analysis was carried out using the statistical package R, version 3.4.3. The study was performed and reported according to the STROBE Statement [18].

### 2.2. Ethical Issues

The research used retrospective anonymized data from the CHSS. No personal data were used, and all the patients data were encrypted, so that no personal identification would be retrievable or traceable to the original source from the working database.

The study complied with the ethical guidelines of the Declaration of Helsinki. The study was revised and approved by the local ethics committee of the Hospital Universitari Parc Taulí in Sabadell (CEIC 2018/625 on date 15 October 2018). Given the retrospective nature of the study, the fact that no personal data were available, and the impossibility of obtaining informed consent for the whole study population, the ethics committee waived the need for informed consent.

## 3. Results

Baseline characteristics of the patients have been described in detail in a previous article [17].

### 3.1. Treatment Time Trends in the Use of Drugs

The percentage of patients treated with salicylates decreased markedly and continuously from 28.8% in 2011 to 17.1% in 2017 (*p* < 0.001). Systemic corticosteroid treatment has also fallen steadily from 15.8% in 2011 to 13.7% in 2017 (*p* < 0.001). The non-biological immunosuppressive treatments showed a smaller reduction, from 32.9% in 2011 to 29.6% in 2017 (*p* < 0.001). Biological treatments increased from 15.0% in 2011 to 18.7% in 2017 (*p* < 0.001) (Figure 1 and Appendix A).

With regard to the different biological treatments (Figure 2 and Appendix A); infliximab was the most used drug in 2011 with 754 patients (8.2% of all CD patients) rising to 1,358 patients in 2017, although the percentage remained stable (8.2%), Adalimumab use increased from 674 patients in 2011 (7.3%) to 1604 patients in 2017 (9.7%). Few patients were treated with golimumab (a maximum of 29 patients in 2016, 0.2%). Ustekinumab was used off-label from 2011 to 2016 with a maximum of 56 patients (0.4%); its use rose to 127 patients (0.8%) in 2017, when it was licensed for CD treatment. Vedolizumab was used first in 2015, when it was administered to 25 patients (0.2%), and its use augmented to 174 patients (1.1%) in 2017.

### 3.2. Surgical Procedures

The absolute number of ostomies and resections increased slightly from 2011 to 2017, from 113 to 152 and from 206 to 279, respectively (Appendix A). However, as the prevalence of CD in this period of time increased markedly, the rate per 1000 patients/year decreased significantly for ostomies (from 13.2 in 2011 to 9.8 in 2017 (*p* = 0.003)), and resections (from 24.1 in 2011 to 18.0 in 2017 (*p* < 0.001)) (Figure 3).

The other surgical procedures showed a similar trend. The absolute number increased from 182 in 2011 to 244 in 2017; the rate per 1000 patients/year, however, declined from 21.3 in 2011 to 15.7 in 2017 (*p* < 0.001) (Figure 3).

There was a positive correlation between the use of salicylates, systemic corticosteroids and non-biological immunosuppressive treatments and surgical procedures. By contrast; biological agents were negatively correlated with surgical procedures. All correlations (except those between corticosteroids and bowel resection) reached statistical significance with a *p* value lower than 0.05 (Appendix A).

### 3.3. Hospital Admissions

The number of CD patients requiring hospitalization due to any cause rose from 2334 in 2011 to 4520 in 2017. Unscheduled hospitalization due to any cause increased from 1316 cases in 2011 to 2560 cases in 2017. Corresponding rates per 1000 patients/year also increased slightly from 272.8 in 2011 to 291.6 in 2017 (*p* < 0.001) and from 153.8 in 2011 to 165.1 in 2017 (*p* < 0.001), respectively (Appendix A).

Regarding specific causes of this hospitalization; CD-related hospitalization decreased from 92.7 per 1000 patients/year in 2011 to 72.2 per 1000 patients/year in 2017 (*p* < 0.001). The rate of infection-related hospitalization rose from 10.9 to 16.1 (*p* < 0.001). Finally, the rate of hospitalization due to cancer remained relatively stable, with slight variations over the period from 17.8 to 19.0, and a maximum value of 22 in 2016 (*p* = 0.02 for the trend) (Appendix A).

Rates of CD-related hospitalization with respect to the general Catalan population increased from 10.56 per 100,000 inhabitants/year in 2011 to 14.8 per 100,000 inhabitants/year in 2017 (*p* < 0.001) (Appendix A).

With regard to the correlation with different treatments; there was a positive correlation between CD-related hospitalization and use of salicylates, corticosteroids and non-biological immunosuppressive treatments. The use of biological treatment showed a negative correlation (Appendix A).

## 4. Discussion

The present study shows that the use of biological drugs increased steadily between 2011 and 2017 in CD patients in Catalonia. Despite this increase, however, only 18.7% of patients were receiving these drugs in 2017. By contrast, the use of salicylates decreased markedly, and treatment with corticosteroids and non-biological immunosuppressive treatments also fell slightly. These changes in drug use correlated with a progressive decrease in the rate of surgeries. Rates of hospitalization due to CD complications also decreased. Therefore, our study suggests that the increase in the use of biological drugs was associated to a reduction in the need for surgery and hospitalization due to complications, although causality cannot be concluded. These data from a large population-based health register of clinical practice, along with the data recently published by Rahman A, et al. [19] in a population-based study from Ontario between 2003 and 2014 (Canada), corroborate previous data obtained in the clinical trials reported above [7,8,9,10].

The observed prevalence of the use of biological drugs (18.7%) is lower than those reported in other western countries. Yu et al. observed a far greater increase in the use of biologic treatment in CD (from 21.8% to 43.8%) in a recent study performed in the United States (US) between 2007 and 2015 [20]. Regarding western European studies, a population-based study in Denmark showed that 23.5% of CD patients were receiving biological treatment in 2012 [11]; in Norway, the use of anti-TNF in CD patients ranged from 20.9% to 31.4% in 2012 [21]; and a French study in 2014 reported rates of anti-TNF use of 33.8% in monotherapy, and 18.3% in combination with a non-biological immunosuppressive treatment [22]. Eastern European figures were much lower; for example, a Hungarian population-based study between 2011 and 2013 reported that only 8.5% of CD patients were receiving biological treatment [23]. These variable rates are probably related to differences in healthcare policies between countries, and highlight the fact that the optimal rates of biological prescription remain unclear. In a prospective population-based study, Vegh et al. showed that surgery and hospitalization rates were much higher in eastern European countries than in western Europe in 2011 [24]. According to our data, it could be hypothesized that this fact might be due, at least in part, to the differences in the use of biological drugs.

Nevertheless, the trends of surgical requirements after the introduction of biological treatment are still unclear. In our study, the rates of surgery procedures fell steadily. Both the ostomy rate and the rate of intestinal resections decreased by 25% between 2011 and 2017: from 13.2 to 9.8 per 1000 patients/year and from 24.1 to 18 per 1000 patients/year, respectively. These rates are similar to those reported by other studies, suggesting that surgical rates in IBD have decreased since the introduction of biological treatment [11,12,25,26,27,28]. In 2013, a meta-analysis of population-based studies demonstrated a reduction of around 25% in surgery rates for CD over the last six decades, from 16.3% and 33.3% at the 1st and 5th years after diagnosis in 1995 to 12.4% and 24.5% in 2000 [29]. More recent retrospective population-based studies have also revealed falls in surgery rates, with reductions of 27% in Poland between 2012 and 2014 [13], and 30% in Québec between 1996 and 2007 [30]. However, the data are heterogeneous and other studies reported that the rate of surgeries remained unchanged or even increased despite biological treatment [31]; in a retrospective study in the US, Lazarev et al. reported that the annual rate of small bowel resection did not change (1.6% between 1995 and 1998 vs. 1.9% between 1999 and 2001 vs. 1.6% between 2002 and 2004 vs. 1.9% between 2004 and 2007) [32]. In a recent retrospective study in Québec, Verdon et al. demonstrated that surgery in CD had risen from 8% in 2010 to 15% in 2015 [14]. Finally, some studies suggest that, although overall surgery rates were decreasing, this was not the case of all types of surgery; Malarcher et al., for instance, showed a reduction in small bowel resection (from 4.9% in 2003 to 3.9% in 2013) whereas colorectal resection and fistula surgery remained stable [33].

Although the rate of CD-related hospitalizations decreased from 92.7 per 1000 patients/year in 2011 to 72.2 in 2017, the total hospitalization rates showed a slight increase. In addition, annual rates per 100,000 inhabitants of CD-related hospitalization rose from 10.6 in 2011 to 14.08 per 100,000 inhabitants/year in 2017, mainly due to the near doubling of the prevalence of the disease [17]. These figures are lower than those presented by King et al. in a worldwide study using the OECD database (Organization for Economic Co-operation and Development), which reported an increase for CD-related hospitalization in Spain with an average annual rate of 23.8 per 100,000 inhabitants/year in the years 2010 to 2015 [34]. Possible explanations for the discrepancy include interregional differences, or differences in the ways hospitalization is reported in Spain’s regional health systems.

Variability in the reported trends and rates of hospitalization in CD occur also in other settings, with significant geographical variations. In a European study based on the European Crohn’s and Colitis Organization EpiCom cohort, Burish et al. demonstrated that 20% of CD patients required CD-related hospitalization in western Europe and 16% in eastern Europe in 2010 [35]. Recently, Murthy et al. showed in a Canadian population-based study between 1995 and 2012 that the use of infliximab did not reduce hospitalization rates (OR 1.06, 95% CI 0.811 to 1.39); they explain that this results from a misguided use of this drug in CD patients [36]. Other studies have recorded an increase in hospitalization rates. In a prospective population-based cohort from Hungary between 2000 and 2010, Golovics et al. observed high rates of CD-related hospitalization, especially during the first-year after the diagnosis, with a probability of 13.6% at one-year follow-up, 23.9% at three years, and 29.8% at five years [37]. Additionally, Sonnenberg et al. found an increase in the hospitalization trend between 1970 and 2004, and attributed this increase to the progressive population aging, with an increase in hospitalization of CD patients in the over-65 age group [15].

Regarding the causes of hospitalization, we observed a 48% increase in the rates of hospitalization due to infectious disease from 2011 to 2017. It is probable that the overall increase in hospitalization is attributable to the rise in infections. This increase might be explained by the use of non-biological immunosuppressive treatments and biological treatments, especially in the elderly population [38]. Shen et al. reported a 4-year infection rate of 43% in elderly vs. 31.6% in young patients receiving biological treatment. Furthermore, the rate of hospitalization due to anti-TNF side effects was 25% in the elderly, compared to 0% in younger patients [39]. We have not analyzed the effect of age on each of the individual causes of hospital admission.

The study has some limitations. First, the CHSS database includes only patients who use the public health system; conceivably, some patients with mild disease might not have been recorded. However, the public system is used by over 80% of the population of Catalonia. The rate of use is therefore much higher, as the 20% of non-users also includes healthy individuals. Furthermore, as CD is a chronic disease and pharmacological (and especially biological) treatments are relatively expensive, most patients request financial support from the public system, which offers universal coverage: in very expensive treatments such as biologics, public prescription approaches 100%.

The data provided by this study and by previous research provides insight into epidemiology and health outcomes of CD that will be useful for future care organization. In addition, the results suggest that biologic treatments may improve outcomes at population levels. As stated, the rate of biological drug use in this study is lower than in other western European countries. The outcomes, however, do not seem to be worse. Although the data of this study showed a low (and decreasing) rate of negative outcomes despite moderate use of biological treatment, the ideal prescription rate for these drugs, in relation to the best achievable outcomes, remains unknown. Future studies should address the heterogeneity in the indication of these drugs and its effect on outcomes.

In conclusion, our study shows that around 20% of CD patients in our country currently receive biological drugs and that this proportion is increasing at a rate of approximately 1% per year. Although causality cannot be proven, this change correlates with a significant decrease in the rates of surgery and CD-related hospitalizations.

## Figures and Tables

**Figure 1 jcm-09-02896-f001:**
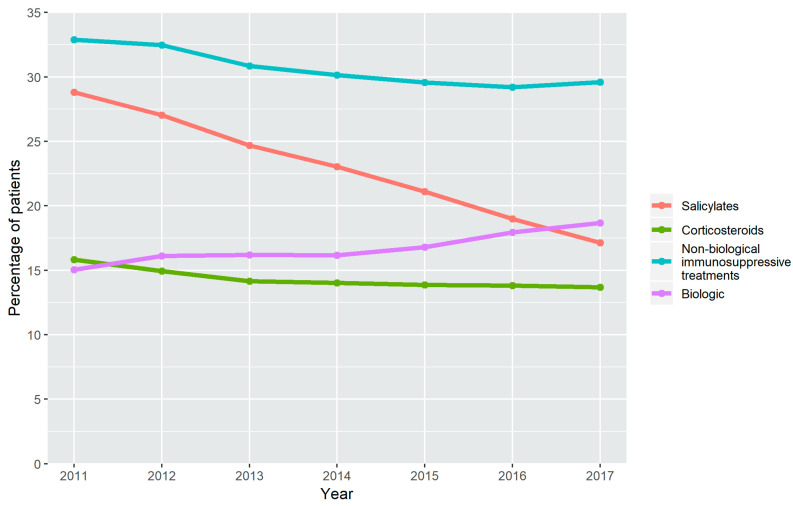
Treatment time trend. Percentage of the treatments used for CD from 2011 to 2017.

**Figure 2 jcm-09-02896-f002:**
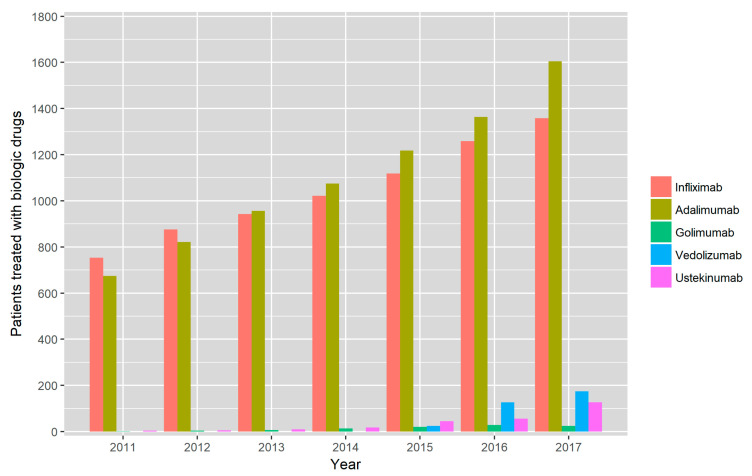
Absolute number of patients treated with biologic drugs from 2011 to 2017.

**Figure 3 jcm-09-02896-f003:**
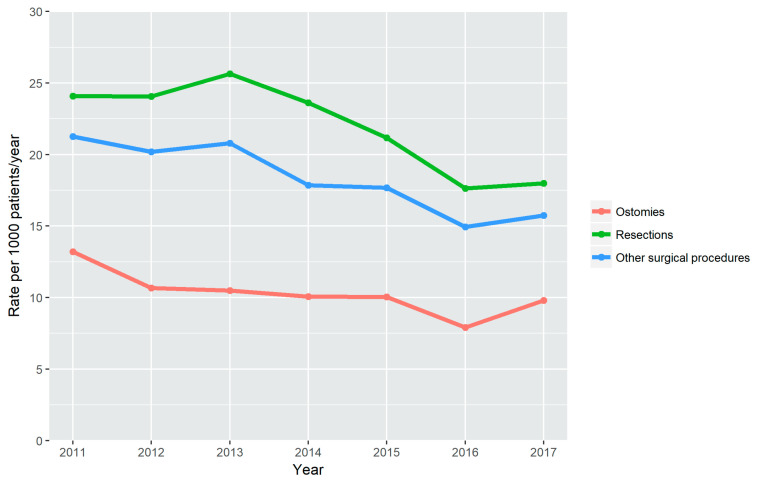
Surgical procedures. Rate per 1000 patients/year from 2011 to 2017.

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
