# Peer review of "Time Trends of Crohn’s Disease in Catalonia from 2011 to 2017. Increasing Use of Biologics Correlates with a Reduced Need for Surgery"

_jcm, 2020, doi:10.3390/jcm9092896_

Round 1
Reviewer 1 Report
Dear editor,
Authors Brunet et al., present this interesting retrospective population-based study looking at the time trends of Crohn’s disease (CD) in Catalonia from 2011 to 2017. Investigators looked up all patients with CD included in the Catalan Health Surveillance System from 2011 to 2017 and retrieved details on exposures to different treatments.
Authors found that between 2011-2017, the use of salicylates, corticosteroids and immunosuppressive treatment fell from 28.8% to 17.1%, 15.8% to 13.7% and 32.9% to 29.6% respectively (p,0.001). On the other hand, the use of biologics use rose from 15% to 18.7% (p,0.001). Ostomy rates per 1000 patients/ year fell from 13.2 in 2011 to 9.8 in 2017 (p=0.003), and surgical resection rates from 24.1 to 18 (p<0.001). The rate of CD-related hospitalizations per 100 patients/ year also fell, from 92.7% to 72.2 (p<0.001). Based on their observations, authors concluded that biological use rose from 15% to 18.7% between 2011 and 2017, and they observed an improvement in the outcomes of CD patients.
Please see my specific comments below:
Abstract: well written, no comments.
Introduction: well written, no comments.
Materials & Methods:
Page 5, line 89: spell check on ‘all-cause’ hospitalization
Results:
Well written, no comments.
Authors mention that the number of Hospital admissions due to any cause in CD patients rose from 2334 to 4520. However, the CD-related hospitalization fell from 92.7 per 100 patient/ year in 2011 to 72.2 per 100 patients/ year in 2017 (p<0.001). The rate of infection-related hospitalization rose from 10.9 to 16.1 (p<0.001). Does this mean the overall increase in hospitalization is due to infections in CD patients? And could this be related to the increased use of biologics?
Discussion:
Well written, no comments.
Maybe the authors can include the answer to my above comment to the discussion section.
Author Response
Thank you for the positive comments
Materials & Methods: Page 5, line 89: spell check on ‘all-cause’ hospitalization
The spell check on page 5, line 89 was reviewed and corrected.
Results and Discussion:
Authors mention that the number of Hospital admissions due to any cause in CD patients rose from 2334 to 4520. However, the CD-related hospitalization fell from 92.7 per 100 patient/ year in 2011 to 72.2 per 100 patients/ year in 2017 (p<0.001). The rate of infection-related hospitalization rose from 10.9 to 16.1 (p<0.001). Does this mean the overall increase in hospitalization is due to infections in CD patients? And could this be related to the increased use of biologics?
Maybe the authors can include the answer to my above comment to the discussion section.
Totally agree. The overall increase of hospitalization is due to infections in CD patients. We tried to discuss this results in lines 273-276. We modified this section for a better compression.
Reviewer 2 Report
Dear authors,
First, I would like to congratulate you for writing an interesting article. The study analyses a large database from Catalonia which provides an insight to the "real-life experience" on biologics use over time and hospitalization-surgical outcomes.
Language
The article is easy to ready, however some grammar/punctuation changes could be performed for better fluency, with special attention in the results section.
Introduction
Lines 59-62: I would recommend not going into to much details on regards to prior studies. Short, concise sentences conveying the message are better.
Methods
It would be worth to mention whether patients in the CHSS database have a unique personal identificator. Given the database registers records information on the care provided at all public health centers in Catalonia, it would be good to know if multiple registrations of the same patient may occur.
Results
Line 144-145: percentage 0.8% is mentioned twice.
I wish the authors continued success.
Author Response
Thank you for the positive comments
Language: The article is easy to ready, however some grammar/punctuation changes could be performed for better fluency, with special attention in the results section.
English language was revised by a native English expert in medical writing. The results section was reviewed again by him.
Introduction: Lines 59-62: I would recommend not going into too much details on regards to prior studies. Short, concise sentences conveying the message are better.
The introductory part of the previous studies has been changed, simplifying the sentences and data given above. Lines 59-63
Methods: It would be worth to mention whether patients in the CHSS database have a unique personal identificator. Given the database registers records information on the care provided at all public health centers in Catalonia, it would be good to know if multiple registrations of the same patient may occur.
CHSS database have a unique and personal identification given by the national health insurance number, which is like the national identity form. This information was added to the manuscript on line 76-78.
Results: Line 144-145: percentage 0.8% is mentioned twice.
The repeated percentage was erased.